# Green Land Use Efficiency and Influencing Factors of Resource-Based Cities in the Yellow River Basin under Carbon Emission Constraints

**Meijing Chen** [1,2], **Qingri Wang** [2,*], **Zhongke Bai** [1,3,4], **Zeyu Shi** [1], **Peng Meng** [2] and **Miao Hao** [2]

[1] School of Land Science and Technology, China University of Geosciences (Beijing), Beijing 100083, China; 3012200009@cugb.edu.cn (M.C.); baizk@cugb.edu.cn (Z.B.); 3012200007@cugb.edu.cn (Z.S.)

[2] China Land Surveying and Planning Institute, Beijing 100035, China; mengpeng@mail.clspi.org.cn (P.M.); haomiao@mail.clspi.org.cn (M.H.)

[3] Key Laboratory of Land Consolidation and Rehabilitation, Ministry of Natural Resources, Beijing 100035, China

[4] Technology Innovation Center of Ecological Restoration Engineering in Mining Area, Ministry of Natural Resources, Beijing 100083, China

[*] Correspondence: wangqingri@mail.clspi.org.cn

**Abstract:** Green and low-carbon strategies represent governance orientations for resource-based cities to respond to global changes and achieve sustainable development. Designating the Yellow River Basin (YRB), an important ecological functional area and an ecologically fragile area, as the research area, this study used the super-efficiency SBM model while considering undesirable outputs, including carbon emissions, to analyze green land use efficiency (GLUE) and its temporal and spatial differentiation, then used the Tobit regression model to analyze the influencing factors. The results were as follows: (1) The GLUE of the YRB presented a spatial pattern of "high in the west and low in the east". (2) Overall, the efficiency values of all areas and types increased annually, but differences occurred in various areas and types of resource-based cities. (3) Overall, the efficiency values of the Yellow River Basin showed a "high-low" polarization. (4) Economic development and population growth factors substantially impacted the GLUE of resource-based cities in this region. It is concluded that increasing the efficiency improvement of low-efficiency regions or cities can improve regional GLUE. To ensure regional green and low-carbon transformation and development, it is essential to enhance urban economic vitality, promote an orderly population flow, and strive to improve social and public services.

**Keywords:** resource-based city; green land use efficiency (GLUE); carbon emissions; the Yellow River Basin (YRB)

## 1. Introduction

As a space carrier for the exchange and interaction of all urban elements, urban land plays an important basic role in developing cities and even regions [1,2]. Although urban land area accounts for only 2% of the global land area, urban energy consumption and carbon emissions account for 75% and 78%, respectively [3], the most important carbon sources in the global terrestrial ecosystem. With the development of urbanization and industrialization, the population continues to gather in cities. To maintain the normal operation of production and life, coupled with people's pursuit of a high-quality lifestyle, under the current energy consumption structure and land use mode, cities will consume more and more energy sources and will continue to emit more and more carbon dioxide into the atmosphere. The resulting climate warming will damage the global ecological environment [4,5]. As the country with the largest carbon emission in the world [6] from 2005 to 2015, China's urban built-up area increased from 32,221 km² to 52,102.31 km², and the urban resident population increased from 562.1 million to 7711.6 million [7]. Therefore,

China has also become an important participant in global climate governance. To cope with the crisis brought about by global climate change, the Chinese government has put forward the double carbon goal, which requires achieving a "carbon peak" by 2030 and "carbon neutralization" by 2060. Various industries and regions have carried out many theoretical and practical explorations of green and low-carbon transformation and Development [8–11]. Resource-based cities have provided important energy and resource guarantees for China's economic and social development and promoted the process of industrialization and urbanization. However, as the industrial systems of resource-based cities focus on resource extraction, transportation, and processing, they have the characteristics of high pollution and high energy consumption. With the gradual depletion of regional resources in resource-based cities, it is vital to explore what is needed to ensure transformation and development. Based on the dual "carbon peak" and "carbon neutralization" visions, resource-based cities must be guided by green and low-carbon governance, upgrade urban industrial systems, and achieve sustainable development. In recent years, national and international scholars have thoroughly examined the transformation obstacles and countermeasures of resource-based cities [12–16], high-quality development [17–22], and land use efficiency [23–27]. However, research on green transformations and the development of resource-based cities [28,29] is still insufficient, especially regarding potential green and low-carbon governance strategies of resource-based cities in combination with the carbon source/sink effect. Therefore, this study examines the green land use efficiency (GLUE) of resource-based cities under the constraint of treating carbon emissions as an undesirable output factor. Using the resource-based cities in the Yellow River Basin (YRB) as an example, this study investigates their GLUE, temporal and spatial differentiation characteristics, and evolution characteristics. Moreover, we further explore the influencing factors and action mechanisms of these characteristics and provide a decision-making basis for promoting the green and low-carbon transformation and development of resource-based cities.

## 2. Materials and Methods

### 2.1. Study Area

The Yellow River is the second-largest river in China. It originates from the northern foot of the Bayankala Mountains on the Qinghai Tibet Plateau, flows through the nine provinces and autonomous regions of Qinghai, Sichuan, Gansu, Ningxia, Inner Mongolia, Shanxi, Shaanxi, Henan, and Shandong, and enters the Bohai Sea in Kenli County, Shandong Province [30]. It has a total length of 5464 km and a drainage area of 795,000 km$^2$, and it is located between 32–42° N and 96–119° E (Figure 1). The YRB is the birthplace of Chinese civilization and is an important area for population activities and economic development. This region has rich reserves of coal, oil, natural gas, and nonferrous metals. In addition to its importance as an energy, chemical, raw material, and basic industrial base in China, it also acts as an important ecological functional area. This is an important part of China's "Two Screens and Three Belts" strategic ecological security pattern. However, the natural conditions in this area are poor, with many practical problems such as shortages of water resources, serious soil erosion, a weak carrying capacity of the resources and environment, and unbalanced and insufficient development of its various subregions. In response, the state has issued an outline of the plan for *the Ecological Protection and High-quality Development of the Yellow River Basin*. Combined with *the National Sustainable Development Plan for Resource-based Cities (2013–2020)*, the YRB includes 38 resource-based prefecture-level cities. According to the data acquisition and changes in administrative divisions, we selected 35 resource-based cities, except Aba in Sichuan, Haixi in Qinghai, and Laiwu in Shandong, as our research objects. This list includes eight cities in the upstream area, 21 cities in the midstream area, and six cities in the downstream area.

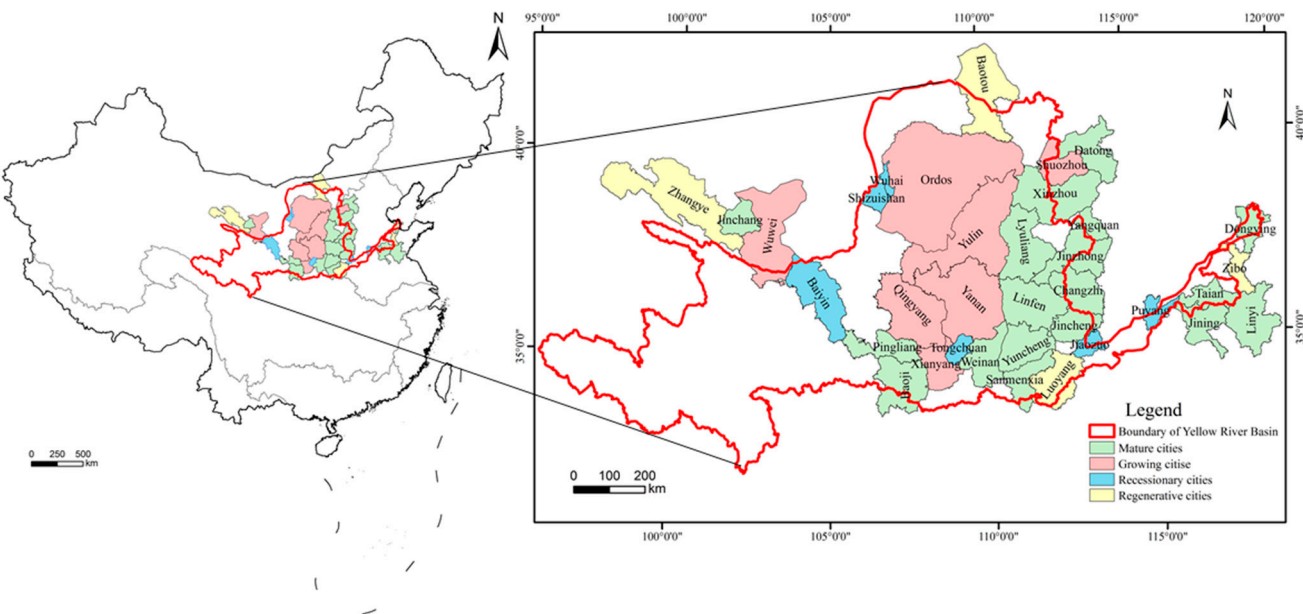

**Figure 1.** Location of resource-based cities in the YRB.

### 2.2. Data Sources

The data used in this study were obtained from the *China Urban Statistical Yearbook*, *China Urban Construction Statistical Yearbook,* and *China Energy Statistical Yearbook* from 2005 to 2020. Some data were supplemented by statistical yearbook data of the corresponding year of the city, and missing data were supplemented by interpolation.

### 2.3. Methods

First, the super-efficiency SBM model considering undesirable outputs was used to calculate the GLUE of resource-based cities in the YRB from 2004 to 2019, and its spatial visualization was expressed using the ArcGIS 10.6 software. Second, the nonparametric Kernel Density Estimation method was used to describe the evolution trend of GLUE in the Yellow River Basin, its various areas, and differing types of resource-based cities. Finally, the Tobit regression model was used to analyze the influencing factors affecting the GLUE of resource-based cities in the YRB (Figure 2).

#### 2.3.1. Super-Efficiency SBM Model Considering Undesirable Output

To scientifically evaluate the relative effectiveness of decision-making units in the case of multiple inputs and outputs, researchers have mostly used the nonparametric DEA method to evaluate land use efficiency [31–34]. With continuous advancements in ecological civilization, the concept of green development has gradually encompassed all aspects of economic and social development. An evaluation of land use efficiency must therefore consider economic and social output indicators, environmental pollution, and greenhouse gas emissions caused by development and construction. Therefore, green land use has become a popular topic in land science research [35–42]. Tone proposed the SBM model based on undesirable outputs [43], which can specifically address the undesirable output problems attached to land use. Moreover, it can accurately reflect the comprehensive realization degree of land use in multiple dimensions such as economic growth, social welfare improvement, resource conservation, pollution, and emission control [44]; however, it still cannot solve the decomposition problem of the efficiency value of effective decision-making units (the efficiency value is 1). Therefore, based on the traditional SBM model, Tone proposed a super-efficiency SBM model considering undesirable outputs [45], which can accurately estimate the super-efficiency value of a decision-making unit and avoid the information loss from an effective decision-making unit. In recent years, this approach has

been widely used in research on green efficiency evaluations of land use [38,44,46–49]. The specific model [50–52] is as follows:

$$\min\rho = \frac{\frac{1}{m}\sum_{i=1}^{m}(\overline{x}/x_{ik})}{\frac{1}{r_1+r_2}\left(\sum_{s=1}^{r_1}\overline{y^d}/y_{sk}^d + \sum_{q=1}^{r_2}\overline{y^u}/y_{qk}^u\right)} \quad (1)$$

$$\begin{cases} \overline{x} \geq \sum_{j=1,\neq k}^{n} x_{ij}\lambda_j; \ \overline{y^d} \leq \sum_{j=1,\neq k}^{n} y_{sj}^d\lambda_j; \ \overline{y^d} \geq \sum_{j=1,\neq k}^{n} y_{qj}^d\lambda_j; \\ \overline{x} \geq x_k; \ \overline{y^d} \leq y_k^d; \overline{y^u} \geq y_k^u; \lambda_j \geq 0; \\ i = 1,2,\cdots,m; j = 1,2,\cdots,n; \\ s = 1,2,\cdots,r_1; q = 1,2,\cdots,r_2 \end{cases} \quad (2)$$

$n$ decision-making units are assumed, and each decision-making unit is composed of input $m$, desirable output $r_1$ and undesirable output $r_2$. $x$, $y^d$, and $y^u$ are the elements in the corresponding input matrix, desirable output matrix, and undesirable output matrix, respectively, and $\rho$ is the value of GLUE in resource-based cities.

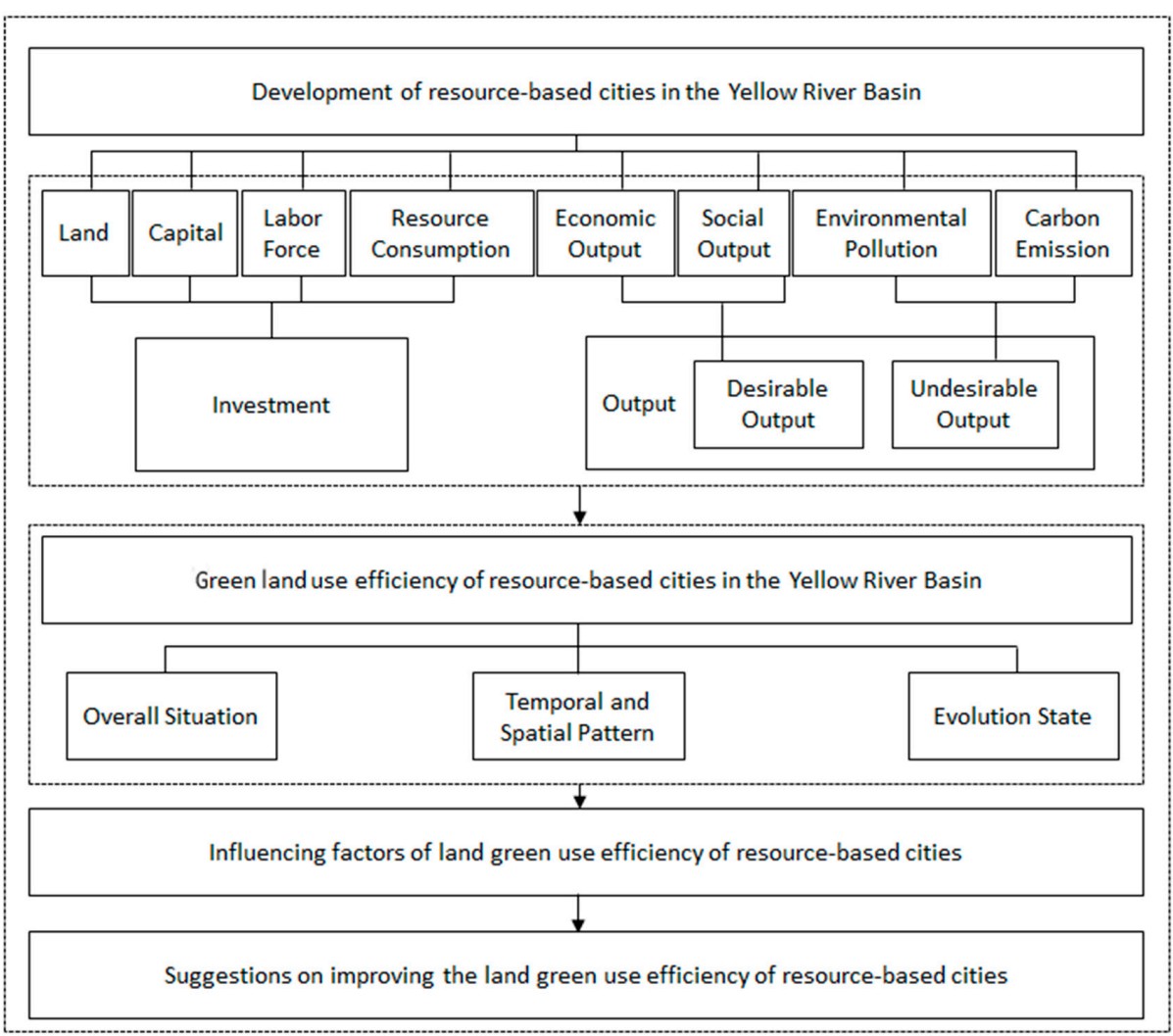

**Figure 2.** Research technology methodology.

### 2.3.2. Nonparametric Kernel Density Estimation

Kernel Density Estimation (KDE) is a nonparametric method for estimating the probability density function. The center of gravity position of the nuclear density curve reflects the evolution characteristics of the size of the efficiency value. The number and height of the peaks can detail the polarization and differential evolution characteristics of the efficiency. The tailing situation can describe the evolution characteristics of the efficiency value in the high/low-value region. The specific model [53] is as follows:

$$f(x) = \frac{1}{Nh} \sum_{i=1}^{n} K\left(\frac{x_i - \overline{x}}{h}\right) \tag{3}$$

where $n$ is the number of samples, $h$ is the window width, $x_i$ is the sample observation value, $K(\cdot)$ is the random kernel function, and $\overline{x}$ is the mean. The type of random kernel function is the Epanechnikov Kernel.

### 2.3.3. Tobit Regression Model

The GLUE calculated in this study was greater than zero and represented a limited dependent variable. Therefore, the Tobit model was used to analyze the factors influencing GLUE each year. The specific model construction [49] is as follows:

$$y_{it} = \begin{cases} y_{it}^* = \beta_0 + \sum_{j=1}^{8} \beta_j X_{j,it} + e_{it}, \ y_{it}^* > 0 \\ 0, y_{it}^* \leq 0 \end{cases} \tag{4}$$

where $y$ is the value of GLUE, $i$ is a resource-based city, $t$ is the year, $x_j$ is the independent variable, $\beta_0$ is the constant term, $\beta_j$ is the vector of the regression coefficient of the independent variable, and $e_{it}$ is the random error term in the regression obeying $N(0, \sigma^2)$.

### 2.4. Index Determination and Data Processing

### 2.4.1. Evaluation of GLUE in Resource-Based Cities

The evaluation indicators selected in this study were combined with the characteristics of resource-based cities [54]. During the selection of input indicators, in addition to the traditional factors of land, capital, and labor, resource consumption was also included in the cost. Regarding undesirable output indicators, in addition to the emissions of industrial wastewater, waste gas, and smoke and dust, the carbon emission indicator was included in combination with the goal of "carbon neutralization" (Table 1).

**Table 1.** Evaluation index system of GLUE in resource-based cities.

| Indicator Type | Considerations | Specific Indicators | Unit |
|---|---|---|---|
| Input | Land Input | Area of urban construction land in municipal districts | $km^2$ |
| | Capital Input | Investment in fixed assets in municipal districts | $10^4$ yuan |
| | Labor Input | Employees in secondary and tertiary industries in municipal districts | person |
| | Resource Consumption | Total urban water supply | $10^4$ $m^3$ |
| | | Total electricity consumption in municipal districts | $10^4$ kWh |
| Desirable Output | Economic Output | GDP in municipal districts | $10^4$ yuan |
| | Social Output | Average wage of on-the-job employees in municipal districts | yuan |
| Undesirable Output | Environmental Pollution | Industrial sulfur dioxide emissions | t |
| | | Industrial smoke and dust emission | t |
| | | Industrial wastewater discharge | $10^4$ t |
| | Carbon Emission | Carbon emission of energy consumption and household | t |

Urban carbon dioxide emissions are mainly generated during the process of energy consumption and residents' lives. Among them, the calculation formula for energy consumption and carbon emissions is:

$$E_1 = \sum_{i=1}^{n} M_i = \sum_{i=1}^{n} e_i \times \alpha_i \times \beta_i \tag{5}$$

where $E_1$ is the carbon emission from energy consumption, $M_i$ is the carbon emission generated by the $i$th energy consumption, $e_i$ is the consumption of class $i$ energy, $\alpha_i$ is the conversion coefficient of standard coal for class $i$ energy, and $\beta_i$ is the carbon emission coefficient of class $i$ energy. The conversion coefficient and carbon emission coefficient of various energy standard coals were obtained from relevant research [55–58] and the IPCC national greenhouse gas inventory guidelines [59] (Table 2). Because the obtained energy consumption data are counted using provincial administrative units, this study designates the energy consumption ratio of each province/autonomous region to the number of prefecture-level cities under its jurisdiction as the energy consumption of each resource-based city.

**Table 2.** Carbon emission coefficients of energy.

| Types of Energy | Conversion Coefficient of Standard Coal | Coefficient of Carbon Emission (kg/kgce) | Types of Energy | Conversion Coefficient of Standard Coal | Coefficient of Carbon Emission (kg/kgce) |
|---|---|---|---|---|---|
| Coal | 0.714,3 | 0.755,9 | Diesel Oil | 1.457,1 | 0.592,1 |
| Coke | 0.971,4 | 0.855,0 | Fuel Oil | 1.428,6 | 0.618,5 |
| Crude Oil | 1.428,6 | 0.585,7 | Liquefied petroleum Gas | 1.714,3 | 0.504,2 |
| Gasoline | 1.471,4 | 0.553,8 | Natural Gas | 1.214,3 | 0.448,3 |
| Kerosene | 1.471,4 | 0.571,4 | Electric Power | 0.122,9 | 0.733,0 |

Note: in the conversion coefficient of standard coal, the unit of liquefied petroleum gas and natural gas is kgce/m$^3$, the unit of electric power is kgce/kW, and other units are kgce/kg.

Residential carbon emissions denote the sum of carbon emissions generated by all residents in the city in a year, and its estimation formula is:

$$E_2 = p \times q_i \tag{6}$$

where $E_2$ is the total carbon emissions from residents, $p$ is the carbon emission coefficient per capita, which is 79 kg/a in this study with reference to relevant research [55], and $q_i$ is the population of the $i$th city.

The carbon dioxide emissions of each city in each year can be obtained by summing the carbon emission $E_1$ of energy consumption and the carbon emission $E_2$ of residents.

### 2.4.2. Influencing Factors of GLUE in Resource-Based Cities

After combining the characteristics of resource-based cities and comprehensively considering the impact of economic, social, and environmental conditions on the development of resource-based cities, eight factors (population growth, economic development, industrial structure, cultural development, medical conditions, education investment, science and technology investment, and environmental management) were selected to investigate the influencing factors of GLUE of resource-based cities in the YRB (Table 3).

**Table 3.** Index system of influencing factors of GLUE in resource-based cities.

| Variable Name | Influencing Factors | Specific Indicators | Unit |
| --- | --- | --- | --- |
| Peop | Population Growth | Natural population growth rate | % |
| Econ | Economic Development | Per capita GDP | yuan/person |
| Indu | Industrial Structure | Proportion of output value of tertiary industry and secondary industry | % |
| Cult | Cultural Development | Proportion of people with a college degree or above | % |
| Hosp | Medical Conditions | Hospital beds per 10,000 people | bed |
| Educ | Education Investment | Proportion of education expenditure | % |
| Scie | Science and Technology Investment | Proportion of science and technology expenditure | % |
| Envi | Environmental Management | Comprehensive utilization rate of industrial solid waste | % |

## 3. GLUE in Resource-Based Cities in the YRB

From 2004 to 2009, the overall GLUE of the YRB showed a steady growth trend (Figure 3). The average efficiency values were 0.681, 0.761, 0.806, and 0.883 in 2004, 2009, 2014, and 2019, respectively, with a total increase of 29.65% over 15 years.

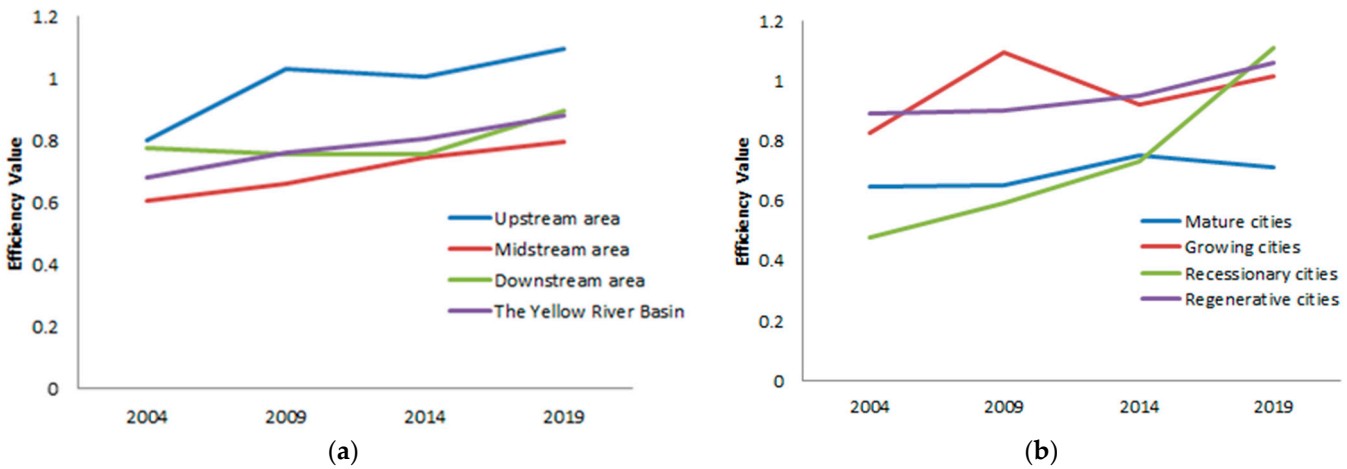

**Figure 3.** GLUE of the resource-based cities in the YRB. (**a**) GLUE of each area; (**b**) GLUE of each type.

### 3.1. Comprehensive Study of GLUE

3.1.1. Characteristics of GLUE of Each Area

There are clear regional differences in the GLUE of resource-based cities in the YRB: (1) Distribution. From 2004 to 2019, the average efficiency of the upstream area was the highest (0.984), followed by the downstream area, with an average efficiency of 0.797, and the average efficiency in the midstream area was the lowest (0.702). (2) Change in trends. During the study period, the efficiency values in the upstream, midstream, and downstream areas showed an overall growth trend. Only the upstream and downstream areas decreased slightly, in 2009–2014 and 2004–2009, respectively, whereas the efficiency values of other regions and periods increased. (3) Changes in range. Over the past 15 years, the growth rates of efficiency values in the upstream and midstream areas exceeded the overall growth rate of the YRB, reaching 36.87% and 31.33%, respectively. This indicates that the upstream and midstream areas actively worked towards a green and low-carbon transformation of industry and strove to achieve high-quality development of land use efficiency; by contrast, the growth rate of efficiency in the downstream area was slow, at only 15.11%. This area will face pressures and challenges in implementing a green land use transformation in the future.

3.1.2. Characteristics of GLUE of Each Type

Significant differences were observed in GLUE among different types of resource-based cities: (1) Distribution. From 2004 to 2019, the average efficiency values of growing

and regenerative cities were large (0.966 and 0.952, respectively), followed by recessionary cities (average efficiency of 0.729) and mature cities (0.692). (2) Change in trends. During the study period, the efficiency value of all types of resource-based cities showed an overall growth trend, except that growth and mature cities decreased in 2009–2014 and 2014–2019, respectively. The efficiency values of other types of resource-based cities increased during each period. (3) Changes in range. Recessionary cities showed the largest increase in efficiency, with an increase of 132.21% over the past 15 years, more than five times that of other types of resource-based cities. The growth rate from 2014 to 2019 was the highest. This showed that recessionary cities completed developmental transformation and gradually eliminated industrial dependence on the path of high resource consumption. The growth rate of the efficiency of growing and regenerative cities was relatively large, at 23.13% and 19.40%, respectively, whereas that of mature cities was only 10.70%. Owing to its low efficiency, mature cities will face a relatively large transformation pressure in future development.

### 3.2. Spatial and Temporal Pattern of GLUE

Using the natural breakpoint method, the GLUE values of resource-based cities in the YRB for each year were divided into five levels in different colors: low, medium-low, medium, medium-high, and high levels. The efficiency value ranges for each level in the different years are shown in Figure 4. Spatially, the GLUE value of resource-based cities in the YRB presented an overall pattern of "high in the west and low in the east", although efficiency rebounds occurred within the extreme eastern areas. Resource-based cities with high-efficiency values were mostly distributed in Gansu, Inner Mongolia, Shaanxi in the upstream area, and Shandong in the downstream area; however, resource-based cities in Shanxi and Henan in the midstream area had low-efficiency values. The efficiency values of Shanxi and Northern Shaanxi in the midstream area decreased significantly during the study period, especially from 2004 to 2009, and the northern region showed a downward trend following continuous improvement. In addition, the efficiency values of the northern cities in the eastern extreme of the YRB were high and rose steadily, whereas those of the southern cities were low and showed a significant downward trend.

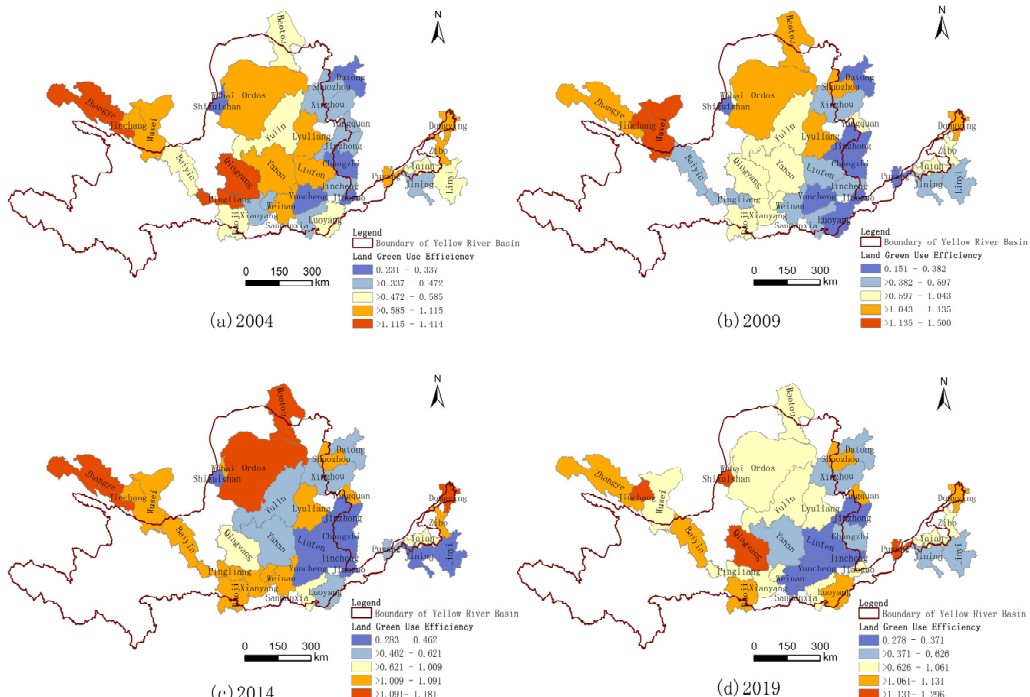

**Figure 4.** Temporal and spatial pattern of GLUE of resource-based cities in the YRB. (**a**) 2004; (**b**) 2009; (**c**) 2014; (**d**) 2019.

### 3.3. Evolution Characteristics of GLUE

In this study, Eviews software (version 10.0) was used to draw the nuclear density curve of GLUE of resource-based cities in the YRB and the temporal dynamic evolution characteristics of GLUE at the global and local scales, respectively (Figure 5). During the study period, the nuclear density curves of the YRB showed typical bimodal characteristics, indicating that the efficiency value of the region showed polarization overall. The peak position of the low-value area on the left moved significantly to the right, whereas the peak position of the high-value area changed minimally. This indicated that the efficiency values of the low-value area significantly improved. It also showed that the improvement of the efficiency of the low-value area comprised the main reason for the increase in the efficiency value of resource-based cities in the YRB. Over time, the left peak gradually decreased and the right peak gradually increased, indicating that the difference in efficiency in low-value areas gradually converged among resource-based cities, whereas the difference in efficiency in high-value areas continued to expand. The tail on the right side of the curve was larger than that on the left. The whole right tail showed the phenomenon of "lengthening-shrinking-lengthening"; this indicated that the efficiency value in the high-value area showed a fluctuating trend of first increasing, then decreasing, and increasing again.

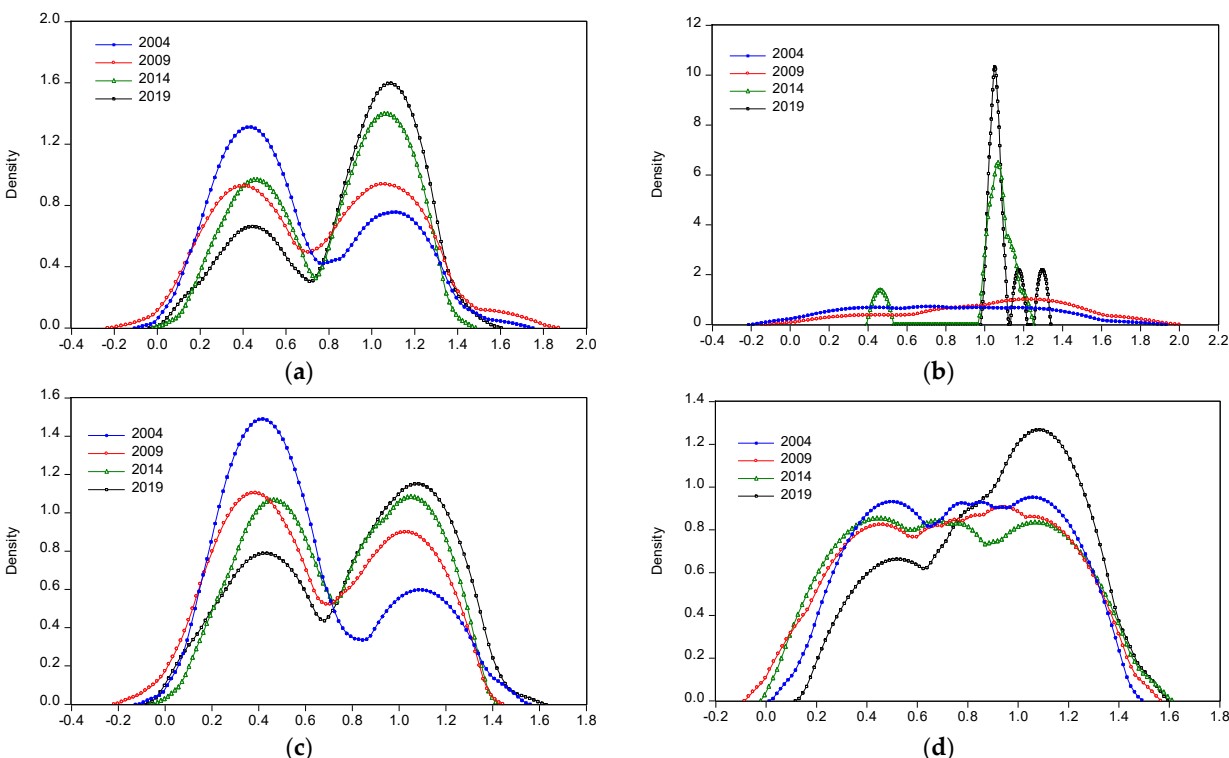

**Figure 5.** Evolution of GLUE of resource-based cities in each area of the YRB. (**a**) The YRB; (**b**) Upstream area; (**c**) Midstream area; (**d**) Downstream area.

3.3.1. Time Series Evolution of Each Area

(1) Upstream area. From 2004 to 2009, the nuclear density curve in the upstream area was flat, and the variation range was not obvious, indicating few differences and changes in the efficiency during this period. From 2014 to 2019, the nuclear density curve showed obvious peaks, and these peaks rose sharply, indicating that the efficiency values in the upstream areas significantly differed during this period, and the differences among cities were rapidly widened. In 2014, there were two peaks, which indicated that the efficiency values during this period showed a strong differentiation in high and low values. Whereas in 2019, there were three peaks, and all were concentrated in the high-value area, indicating that the efficiency values in this period were relatively high, and the differences were obvious.

(2) Midstream area. Owing to the relatively large number of resource-based cities in the midstream areas, this region showed characteristics similar to the overall efficiency evolution of the YRB. The main differences were the variation in the amplitude of the wave crest and the shape of the tail. The degree of contraction of the left peak in the midstream area was higher than that of the YRB, and the uplift degree of the right peak was less than that of the YRB. This indicated that the reduction in the degree of differentiation in the middle reaches was more obvious in the low-value area, whereas the expansion trend of differentiation in the high-value area was smaller than that in the YRB. In the tail of the curve, the right tail shrank first and then grew, thereby indicating that the efficiency value of the high-value area increased after a continuous reduction.

(3) Downstream area. From 2004 to 2014, the variation range of the nuclear density curve in the downstream area was not obvious, and the position of the wave peak showed an overall downward trend. The left peak shifted slightly to the left, whereas the right peak remained relatively stable; this indicated that during this period, the efficiency of resource-based cities in the downstream area changed minimally, the efficiency difference between cities continued to narrow, the efficiency in the low-value area decreased slightly, and the change in the high-value area was not significant. By 2019, the left peak decreased significantly, whereas the right peak rose sharply, and the left peak moved slightly to the right; this indicated that the difference in the efficiency in the low-value area decreased significantly, whereas the difference in the efficiency value in the high-value area expanded rapidly, and the efficiency value in the low-value area increased. During the study period, the left tail was relatively obvious, first reduced and then extended, indicating that the efficiency value in the low-value area first decreased and then increased.

### 3.3.2. Time Series Evolution of Various Types

The GLUE values of different types of resource-based cities showed different evolution laws (Figure 6).

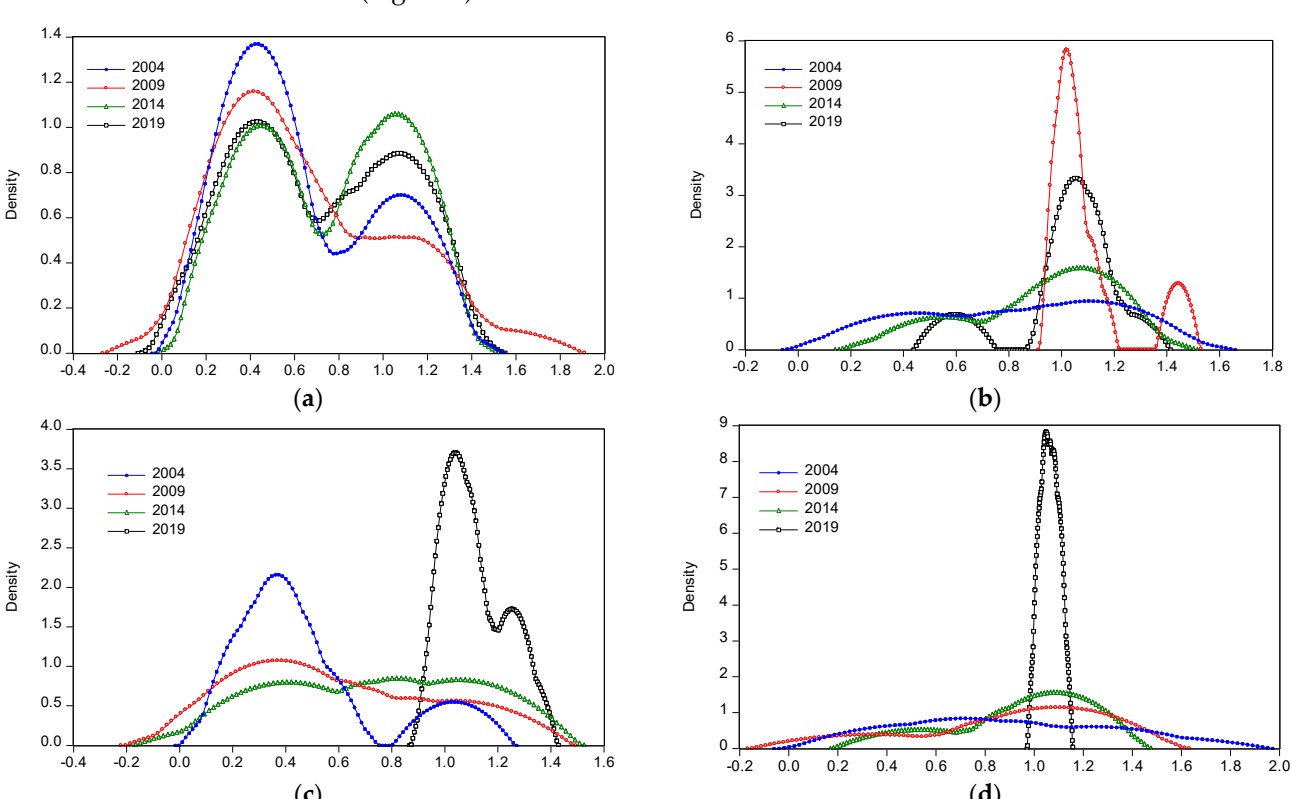

**Figure 6.** Evolution of GLUE of resource-based cities in each type of the YRB. (**a**) Mature cities; (**b**) Growing cities; (**c**) Recessionary cities; (**d**) Regenerative cities.

(1) Mature cities. From 2004 to 2019, the core density curve of the efficiency value of mature cities had a typical bimodal structure, indicating that the efficiency value of mature cities presented the differentiation characteristics of high value and low value. The wave crest in the low value area decreased gradually, while the wave crest in the high value area showed the characteristics of increased volatility, indicating that the difference of efficiency value in the low value area decreased gradually, while the difference in the high value area changed greatly and increased as a whole.

(2) Growing cities. During the study period, the efficiency value curve of growing cities changed greatly. The nuclear density curve in 2004 and 2014 was relatively flat, and the peak was not obvious, indicating that there was little difference in the efficiency value in this period. The annual average for 2009 and 2019 was in the form of double peaks, indicating an obvious differentiation in the efficiency value in this period. In 2009, the two peaks shifted significantly to the right, indicating that the efficiency value of low value area and high value area increased rapidly in this year; The wave crest in the low value area rose first and then decreased, while the wave crest in the high value area continued to rise, indicating that the difference of efficiency value in the low value area changed greatly, expanded first and then tended to converge, while the difference in the high value area increased continuously.

(3) Recessionary cities. In 2009 and 2014, the efficiency value curve of recessionary resource-based cities was relatively flat, while in 2004 and 2019, the curve changed significantly and showed bimodal characteristics, indicating that the efficiency value of this type of city changed from obvious differentiation to convergence, and the evolution characteristics of differentiation had appeared in recent years. Both the left peak and the right peak migrated significantly to the right, and the left peak decreased first and then increased significantly, and the right peak continued to rise, indicating that the efficiency values of the low value area and the high value area had increased, the low value area tended to converge and then expanded the difference, while the difference of the high value area continued to increase.

(4) Regenerative cities. From 2004 to 2014, the efficiency value curve of this type of city was relatively flat, with only a slight bimodal structure in 2014, while in 2019, the change was large, and the single peak was significantly prominent, indicating that the change and difference in efficiency value in the early stage were not obvious, only there were signs of differentiation in the middle stage, while the difference in the later stage was sharply enlarged. The trailing indentation on the right side of the curve was obvious, indicating that the efficiency value in the high value area decreased significantly.

## 4. Analysis of Influencing Factors of GLUE in Resource-Based Cities

Tobit regression analysis of influencing factors of GLUE of resource-based cities in the YRB was performed using the Eviews 10.0 software. The model operation results showed that the log likelihood values in 2004, 2009, 2014, and 2019 were −4.340, −5.436, 1.790, and 1.423, respectively, and the *AIC* values were 0.819, 0.882, 0.469, and 0.490, respectively, thereby indicating that the model fitting effect was satisfactory (Table 4).

Table 4 demonstrates differences in the influence of various variables on the GLUE of resource-based cities. According to the average value of the influence coefficient for each year, the impact of economic development factors was the strongest. For every percentage point increase, the GLUE of urban land increased by 0.724 percentage points. The second strongest factor was population growth; for every percentage point increase, the efficiency value decreased by 0.406 percentage points. By contrast, the influence strength of the medical conditions, cultural level, and science and technology investment factors were weak, and environmental management, industrial structure, and education investment showed no impact whatsoever. (1) Population growth factors. Except for 2009, population growth in other years had a reverse control effect on the value of GLUE, of which 2014 and 2019 had the strongest effect, and both passed the significance level test of 1%. This indicated that excessive population gathering led to excessive consumption of resources, which was not

conducive to facilitating green land use in urban areas. (2) Economic development factors. During the study period, the effect of economic development factors on GLUE was positive, and the regression coefficients passed the significance test. This impact was the strongest in 2009, with a regression coefficient of 1.054; this indicates that, when improving GLUE in resource-based cities, the vitality of urban economic development should be increased, and regional green transformation and development should be promoted through offering economic incentives. (3) Industrial structural factors. The impact of industrial structure on GLUE was negative. This indicates that resource-based cities should constantly optimize their industrial structure during transformation and development, eliminate their dependence on high energy consumption and heavy pollution industries such as mining and manufacturing, and attempt to increase the output value proportion generated by tertiary industry. (4) Cultural development factors. The regression coefficient of the cultural level factor passed the significance level test of 10% in 2014 and 2019, and the action direction was negative; this indicated that improving the cultural level cannot effectively improve GLUE in resource-based cities, but it does provide a certain degree of inhibition. (5) Medical conditions The regression coefficient of medical condition factors passed the significance test in 2004 and 2009, and the action direction was negative. However, this direction was positive in 2014 and 2019, indicating that this factor's impact on the value of GLUE was not stable. (6) Education investment factors. The regression coefficient of this factor was positive in 2004 and 2019 and passed the significance test at the 5% level in 2004; this indicated that education had a relatively obvious and positive role in improving GLUE in resource-based cities. (7) Science and technology investment factors. The regression coefficients of science and technology investment factors were negative, which indicates that excessive science and technology investment inhibited improvements in GLUE within resource-based cities to a certain extent. This result also indicated that the science and technology transformations of resource-based cities in this region did not effectively promote a green transformation and development in land use. (8) Environmental management factors. The effect of environmental management factors on the GLUE was positive, thereby indicating that strengthening environmental management can effectively improve the value of GLUE and promote regional green-and low-carbon transformation.

**Table 4.** Tobit regression results of influencing factors of GLUE on resource-based cities in the YRB.

| Variable Name | 2004 | 2009 | 2014 | 2019 |
|---|---|---|---|---|
| Peop | −0.307 (0.283) | 0.225 (0.240) | −0.854 *** (0.253) | −0.687 *** (0.205) |
| Econ | 0.918 * (0.551) | 1.054 *** (0.292) | 0.410 ** (0.178) | 0.514 ** (0.260) |
| Indu | −0.185 (0.287) | −0.230 (0.235) | −0.066 (0.163) | −0.171 (0.259) |
| Cult | −0.338 (0.277) | 0.059 (0.216) | −0.288 * (0.169) | −0.399 * (0.231) |
| Hosp | −0.437 * (0.248) | −0.958 *** (0.363) | 0.075 (0.212) | 0.328 (0.248) |
| Educ | 0.595 ** (0.302) | −0.312 (0.262) | −0.159 (0.213) | 0.225 (0.272) |
| Scie | −0.401 (0.606) | −0.252 (0.265) | −0.122 (0.157) | −0.098 (0.169) |
| Envi | 0.286 (0.215) | 0.007 (0.200) | 0.283 (0.187) | 0.106 (0.147) |

Note: ***, **, and * indicate significance at the 1%, 5%, and 10% levels, respectively; the figures in brackets are standard errors.

## 5. Discussion

### 5.1. GLUE with Carbon Emissions Included in Undesirable Output

It is of great significance to study the GLUE of resource-based cities under the constraint of carbon emission. With the continuous advancement of ecological civilization construction, the concept of green development has gradually penetrated all aspects of economic and social development. Land use efficiency evaluation needs to consider not only economic and social output indicators but also environmental pollution and greenhouse gas emission caused by development and construction. By comparing the traditional DEA model with the SBM undesirable model considering undesirable output, Zhang found that the latter's efficiency is only 77.04% of the former [60]. It shows that pollution emission

reduces the social welfare benefits of output and has an obvious negative effect on land use efficiency. By studying the GLUE of resource-based cities in the YRB concerning unexpected output (excluding carbon emission), Ding found that the number of cities with an efficiency value of more than 1.06 in 2009 and 2014 was higher than that in this study [54], indicating that when the carbon emission factor is included in the undesirable output for the calculation of GLUE, it will further reduce the social welfare effect of output and lead to the reduction of efficiency value. Therefore, it is very necessary to consider the impact of carbon emissions on the green transformation of resource-based cities under the goal of "carbon peaking" and "carbon neutralization", which can urge local governments to take effective measures to control urban carbon emissions, to help alleviate the crisis caused by global climate change.

### 5.2. GLUE Considering Carbon Emissions and Influencing Factors

By studying the GLUE of resource-based cities under the constraint of carbon emission, it is possible to grasp the changing trend and temporal and spatial differentiation of GLUE of resource-based cities, analyze the reasons for its overall change and spatial difference, and study the action direction and strength of various factors through influencing factor analysis, to provide a basis for local governments to make decisions to improve the GLUE.

(1) The GLUE regarding carbon emissions of resource-based cities in the YRB has increased annually, but the efficiency values of various river basins and types of resource-based cities differed. Among them, the upstream area had the highest efficiency values and the fastest growth rate, whereas the midstream and downstream areas had low efficiency values and low growth rates. The efficiency values of the growing and regenerative resource-based cities were high, whereas those of recessionary and mature resource-based cities were low. However, the fastest growth rate was in recessionary cities, and the growth rates of the other three efficiency types were relatively slow.

(2) The GLUE of resource-based cities in the YRB showed obvious temporal and spatial differences. Overall, the efficiency values presented a spatial pattern of "high in the west and low in the east", but those of some cities in far east Shandong Province increased. During the study period, the upstream area maintained a stable development at a high level, the midstream area decreased significantly, and the northern region decreased following a continuous improvement. The efficiency values of northern cities in the east wing were high and steadily increased, whereas those of the southern cities in the eastern wing were low and decreased significantly.

(3) The GLUE of resource-based cities in the YRB presented the characteristics of "high-low" polarization overall, indicating that the GLUE of resource-based cities in this region presented unbalanced development. During the study period, the low-value areas demonstrated a gradual improvement in efficiency, and the differences tended to converge, whereas the growth of high-value areas was not obvious, and the differences between cities gradually expanded.

(4) Economic development and population growth factors had a substantial impact on the GLUE of resource-based cities in the YRB, followed by the medical conditions, cultural level, and science and technology investment factors. In contrast, environmental management, industrial structure, and education investment factors had minimal impact.

### 5.3. Suggestions for Improving the GLUE

Combined with the above research, some suggestions are put forward to improve the GLUE of resource-based cities in the YRB from the region, type, industry, and factors.

(1) Improve the GLUE of resource-based cities in the midstream and downstream areas to reduce the overall regional difference. Increasing the efficiency improvement of resource-based cities with low efficiency values can effectively improve the overall GLUE value of the region. Therefore, the midstream and downstream areas with low efficiency values should receive extra focus, especially Weinan, Linfen, Yan'an, Jinzhong, Jincheng, and other cities with an obvious decline in efficiency grades in the midstream area. Specifically,

reasons for their low efficiency values and continuous declines should be analyzed, and improvements in urban green and low-carbon governance should be sought.

(2) Promote the GLUE of recessionary and mature cities to reduce the differences between different types of cities. Boost the green and low-carbon transformation of land use in recessionary and mature resource-based cities with low efficiency values, especially the mature cities with the slowest growth rate. Gradually phase out their dependence on industries such as resource mining, processing, and transportation, actively pursue industrial structure reform and transformation, and develop primary and tertiary industries combined with regional characteristics. Examples include exploring the integration of mining and agriculture, developing high-quality agricultural products, integrating mining and tourism, and fostering partnerships with mining schools to build green and low-carbon economic models, such as a science popularization education base.

(3) Adjust the industrial structure and promote the green and low-carbon upgrading of the industrial system of resource-based cities. Respect the life cycle law of resource-based cities, study and judge the resource reserves and exploitation potential of each resource-based city, and realize the reconstruction or upgrading of the industrial system in the early stage of resource depletion in combination with the location conditions and natural endowment characteristics. For resource-based cities without support [61], strive to cultivate and expand the non-mining economy and actively seek to combine with the existing advantageous industries of the city for resource-based cities with support.

(4) Improve and optimize the driving factors to realize the green and low-carbon development of resource-based cities. It is important to improve the urban economic vitality, promote the orderly flow of population, strive to improve social and public services such as medical and cultural conditions, and improve the GLUE of resource-based cities by increasing investment in science, technology and education and strengthening environmental management, especially increasing scientific and technological innovation, strengthen the role of scientific and technological innovation in promoting the upgrading of regional industrial structure and the transformation of green development, to promote the comprehensive, sustainable and high-quality development of the YRB.

## 6. Conclusions

This study considers carbon emissions as an undesirable output factor and evaluates the GLUE of resource-based cities in the YRB, an important ecological functional area in China. It analyzes its temporal and spatial differences and explores its influencing factors and action mechanisms to provide a scientific basis for promoting the green and low-carbon development of the region. Although this study improves the research methods of urban land green use efficiency under the objectives of "carbon peak" and "carbon neutralization", due to the limitation of the refinement of carbon emission data, the scientificity of this study can be further improved in the future. Of course, it is also necessary to further explore how the resource-based cities in the YRB can realize the industrial integrated development of the mine-agriculture-city complex area [62] under the hard constraints of natural conditions, promote the transformation and upgrading of the industrial system to achieve high-quality regional development under the guidance of green and low-carbon governance, and contribute to mitigating the crisis of global climate change.

**Author Contributions:** Conceptualization, M.C. and Z.B.; methodology, Q.W.; software, M.C.; validation, Q.W. and Z.S.; formal analysis, M.C. and P.M.; data curation, Z.B. and M.H.; writing—original draft preparation, M.C.; writing—review and editing, Q.W.; funding acquisition, Z.B. All authors have read and agreed to the published version of the manuscript.

**Funding:** This research was funded by the Natural Resources Investigation, Monitoring and Right Confirmation Registration Project of the Ministry of Natural Resources of China (Project No. JCQQ221602-04).

**Institutional Review Board Statement:** Not applicable.

**Informed Consent Statement:** Not applicable.

**Data Availability Statement:** The data presented in this study are available on request from the corresponding author.

**Conflicts of Interest:** The authors declare no conflict of interest.

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
