# Peer review of "Green Land Use Efficiency and Influencing Factors of Resource-Based Cities in the Yellow River Basin under Carbon Emission Constraints"

_buildings, doi:10.3390/buildings12050551_

Round 1
Reviewer 1 Report
Authors have explained well in the starting what they are aiming for and referred well for the introduction. They have used different methods of statistical, spatial and temporal analysis to present their case and arguments.
I would like to focus on the later part of the paper. I think there should be a sort of discussion section or at least a para, comprehensively providing the result of entire analysis by different methods of Tobit regression, spatial and time based evolution analysis. It may be clear for the authors but not an ordinary reader to comprehend and sum up what all the analysis are pointing to? what is the essence or interpretation of all these results?
Perhaps authors need to look into their four conclusions and try to link them and say what is the importance or why it is important to know about low and high efficiency values in different cities or upstream and downstream? why the polarization of low and high values is important to know and how it may be linked to the spatial and temporal differences of efficiency values? I think a sum up is required to give it more scientific value. The policy recommendations strengthen the paper and I would suggest for the point to mention the role of technology or technological development as a crucial factor in transformation of land use in recessionary and 434 mature resource-based cities. It is implicitly there in this particular recommendation but needs to be mentioned explicitly.
Author Response
1.Authors have explained well in the starting what they are aiming for and referred well for the introduction. They have used different methods of statistical, spatial and temporal analysis to present their case and arguments.
I would like to focus on the later part of the paper. I think there should be a sort of discussion section or at least a para, comprehensively providing the result of entire analysis by different methods of Tobit regression, spatial and time based evolution analysis. It may be clear for the authors but not an ordinary reader to comprehend and sum up what all the analysis are pointing to? what is the essence or interpretation of all these results?
Reply:
Thank you for your suggestion. The discussion section has been added in the manuscript, which expounded in detail the importance and necessity of carrying out this study, and adjusted the relevant contents of the conclusion section to make the format of the manuscript more standardized.
2.Perhaps authors need to look into their four conclusions and try to link them and say what is the importance or why it is important to know about low and high efficiency values in different cities or upstream and downstream? why the polarization of low and high values is important to know and how it may be linked to the spatial and temporal differences of efficiency values? I think a sum up is required to give it more scientific value. The policy recommendations strengthen the paper and I would suggest for the point to mention the role of technology or technological development as a crucial factor in transformation of land use in recessionary and 434 mature resource-based cities. It is implicitly there in this particular recommendation but needs to be mentioned explicitly.
Reply:
We have accepted your suggestions and improved the discussion and conclusion section. In the discussion section, it is pointed out that this study puts forward suggestions to improve the GLUE of resource-based cities in the YRB from the perspectives of region, type, industry and factors, so as to make the correlation between the contents stronger. The explanation of the meaning of the polarization characteristic of "high-low" efficiency value was added, which showed that the GLUE of resource-based cities in this region presented unbalanced development. In the proposed content, the role of technological innovation in promoting the upgrading of regional industrial structure and the transformation of green development had been added.

Reviewer 2 Report
The manuscript titled "Green Land Use Efficiency and Influencing Factors of Resource-based Cities in the Yellow River Basin under Carbon Emission Constraints" intends to explore what is needed to ensure transformation and development, because of the gradual depletion of regional resources in resource-based cities. This study examines the green land use efficiency (GLUE) of resource-based cities under the constraint of treating carbon emissions as an undesirable output factor. Moreover, the manuscript explores the influencing factors and action mechanisms of these characteristics and provide a decision-making basis for promoting the green and low-carbon transformation and development of resource-based cities. This study selects the resource-based cities in the Yellow River Basin in the China as study area.
The research is original; it could be characterized as novel and in my opinion important to the field, it also has almost the appropriate structure and language been used well. In the meanwhile, the manuscript has a short extent (about 5,860 words) and must be more. The tables (4), figures (6) and equations (6) make the paper to reflect well to the reader. For this reason, paper has a "diversity look", not only tables, not only numbers, not only words.
The title is all right. The abstract reflects well the findings of this study, and it has the appropriate length. The introduction is effective, clear, and well organized; it really introduced and put into perspective what research is negotiating but is too short. Please revise the Introduction of the manuscript and include references which are already exists in bibliography (you need to rich about 50 references). Moreover, it does not contain a clear formulation and description of the research problem. This makes it difficult for the reader to understand the argumentation. Please insert a clear description and justification of the problem the article deals with.
Also use the appropriate research manuscript sections: Introduction, Materials and Methods, Results, Discussion and Conclusions, as journal wants (https://www.mdpi.com/journal/buildings/instructions).
For the Methodology chapter, the research conduct has been tested in several areas of the world, with similar results and will probably be tested in others. Appropriate references to the methodology included in the already published bibliography. The methodology followed is sufficiently documented and do not need to be explained clearly. It is advised to revise the Discussion and Conclusion. Both sections should be consistent in terms of Proposal, Problem statement, Results, and of course, future work. Your conclusion section has an appropriate length but does not justice to your work. Make it your key contributions, arguments, and findings clearer. You must refer to the literature and previous studies in your discussion and conclusion sections.
Please revise the references of the manuscript and include references which are already exists in bibliography. I would be much more satisfied if the number of references was slightly higher (about 25 - 30 references) and I would appreciate it if also included data from the entire world (Asia, America, Europe and Australia e.tc.). In this way it is documented that a project which is tested in a place with its own characteristics can be implemented in other places around the world. References must have an appropriate style, for this reason I would be good to change [see: Instructions for Authors / Manuscript Preparation / Back Matter / References: - (https://www.mdpi.com/journal/buildings/instructions or https://www.mdpi.com/authors/references)]. Do not forget, DOI numbers (Digital Object Identifier) are not mandatory but highly encouraged and make the review easier.
Author Response
1.The manuscript titled "Green Land Use Efficiency and Influencing Factors of Resource-based Cities in the Yellow River Basin under Carbon Emission Constraints" intends to explore what is needed to ensure transformation and development, because of the gradual depletion of regional resources in resource-based cities. This study examines the green land use efficiency (GLUE) of resource-based cities under the constraint of treating carbon emissions as an undesirable output factor. Moreover, the manuscript explores the influencing factors and action mechanisms of these characteristics and provide a decision-making basis for promoting the green and low-carbon transformation and development of resource-based cities. This study selects the resource-based cities in the Yellow River Basin in the China as study area.
The research is original; it could be characterized as novel and in my opinion important to the field, it also has almost the appropriate structure and language been used well. In the meanwhile, the manuscript has a short extent (about 5,860 words) and must be more. The tables (4), figures (6) and equations (6) make the paper to reflect well to the reader. For this reason, paper has a "diversity look", not only tables, not only numbers, not only words.
Reply:
Thank you for your suggestion. The content of the manuscript has been expanded to 6640 words, and the relevant expressions have been more perfect and enriched.
2.The title is all right. The abstract reflects well the findings of this study, and it has the appropriate length. The introduction is effective, clear, and well organized; it really introduced and put into perspective what research is negotiating but is too short. Please revise the Introduction of the manuscript and include references which are already exists in bibliography (you need to rich about 50 references). Moreover, it does not contain a clear formulation and description of the research problem. This makes it difficult for the reader to understand the argumentation. Please insert a clear description and justification of the problem the article deals with.
Reply:
We have accepted your suggestions, improved the introduction of the manuscript, re read the relevant literature (the number of references has been increased to 62), elaborated the research background of this study, and highlighted the research significance.
3.Also use the appropriate research manuscript sections: Introduction, Materials and Methods, Results, Discussion and Conclusions, as journal wants (https://www.mdpi.com/journal/buildings/instructions).
Reply:
We have accepted your suggestion, added the discussion section of the manuscript, and adjusted the content of the conclusion section to make the structure of the manuscript more standardized.
4.For the Methodology chapter, the research conduct has been tested in several areas of the world, with similar results and will probably be tested in others. Appropriate references to the methodology included in the already published bibliography. The methodology followed is sufficiently documented and do not need to be explained clearly. It is advised to revise the Discussion and Conclusion. Both sections should be consistent in terms of Proposal, Problem statement, Results, and of course, future work. Your conclusion section has an appropriate length but does not justice to your work. Make it your key contributions, arguments, and findings clearer. You must refer to the literature and previous studies in your discussion and conclusion sections.
Reply:
Thank you for your suggestions. We focused on improving the discussion and conclusion sections of the manuscript. In the discussion section, we described in detail the advantages of this study compared with other relevant studies, as well as the necessity of improving research methods in the context of the national strategy to achieve the goals of "carbon peaking" and "carbon neutralization". And in the conclusion section, we pointed out the improvement direction of relevant research in the future.
5.Please revise the references of the manuscript and include references which are already exists in bibliography. I would be much more satisfied if the number of references was slightly higher (about 25 - 30 references) and I would appreciate it if also included data from the entire world (Asia, America, Europe and Australia e.tc.). In this way it is documented that a project which is tested in a place with its own characteristics can be implemented in other places around the world. References must have an appropriate style, for this reason I would be good to change [see: Instructions for Authors / Manuscript Preparation / Back Matter / References: - (https://www.mdpi.com/journal/buildings/instructions or https://www.mdpi.com/authors/references)]. Do not forget, DOI numbers (Digital Object Identifier) are not mandatory but highly encouraged and make the review easier.
Reply:
We have accepted your suggestion, consulted the literature on green and low-carbon land use research all over the world in more detail, expanded the number of references, and marked the DOI number one by one, so as to enrich, improve and standardize the content of the manuscript.
